# Clinical Disease Severity Mediates the Relationship between Stride Length and Speed and the Risk of Falling in Parkinson’s Disease

**DOI:** 10.3390/jpm12020192

**Published:** 2022-01-31

**Authors:** Yun-Ru Lai, Chia-Yi Lien, Chih-Cheng Huang, Wei-Che Lin, Yueh-Sheng Chen, Chiun-Chieh Yu, Ben-Chung Cheng, Chia-Te Kung, Chien-Feng Kung, Yi-Fang Chiang, Yun-Ting Hung, Hsueh-Wen Chang, Cheng-Hsien Lu

**Affiliations:** 1Department of Neurology, Kaohsiung Chang Gung Memorial Hospital, Chang Gung University College of Medicine, Kaohsiung City 833401, Taiwan; yunrulai@cgmh.org.tw (Y.-R.L.); u9301024@cgmh.org.tw (C.-Y.L.); hjc2828@gmail.com (C.-C.H.); ifmaymay@gmail.com (Y.-F.C.); swear666@cgmh.org.tw (Y.-T.H.); 2Department of Radiology, Kaohsiung Chang Gung Memorial Hospital, Chang Gung University College of Medicine, Kaohsiung City 833401, Taiwan; u64lin@yahoo.com.tw (W.-C.L.); yssamchen@gmail.com (Y.-S.C.); yuchiunchieh@gmail.com (C.-C.Y.); 3Department of Internal Medicine, Kaohsiung Chang Gung Memorial Hospital, Chang Gung University College of Medicine, Kaohsiung City 833, Taiwan; benzmcl@gmail.com; 4Department of Emergency Medicine, Kaohsiung Chang Gung Memorial Hospital, Chang Gung University College of Medicine, Kaohsiung City 833401, Taiwan; kungchiate@gmail.com; 5Department of Intelligent Commerce, National Kaohsiung University of Science and Technology, Kaohsiung 82444, Taiwan; cfkung@nkust.edu.tw; 6Department of Biological Science, National Sun Yat-Sen University, Kaohsiung 80424, Taiwan; hwchang@mail.nsysu.edu.tw; 7Center for Shockwave Medicine and Tissue Engineering, Kaohsiung Chang Gung Memorial Hospital, Chang Gung University College of Medicine, Kaohsiung City 833401, Taiwan; 8Department of Neurology, Xiamen Chang Gung Memorial Hospital, Xiamen 361126, China

**Keywords:** fall risk, Parkinson’s disease, spatiotemporal parameters, three-step fall prediction model

## Abstract

The shuffling gait with slowed speed and reduced stride length has been considered classic clinical features in idiopathic Parkinson’s disease (PD), and the risk of falling increases as the disease progresses. This raises the possibility that clinical disease severity might mediate the relationship between stride length and speed and the risk of falling in patients with PD. Sixty-one patients with PD patients underwent the clinical scores as well as quantitative biomechanical measures during walking cycles before and after dopamine replacement therapy. Mediation analysis tests whether the direct effect of an independent variable (stride length and speed) on a dependent variable (three-step fall prediction model score) can be explained by the indirect influence of the mediating variable (Unified Parkinson’s Disease Rating Scale (UPDRS) total scores). The results demonstrate that decreased stride length, straight walking speed, and turning speed is associated with increased three-step fall prediction model score (r = −0.583, *p* < 0.0001, r = −0.519, *p* < 0.0001, and r = −0.462, *p* < 0.0001, respectively). We further discovered that UPDRS total scores value is negatively correlated with stride length, straight walking, and turning speed (r = −0.651, *p* < 0.0001, r = −0.555, *p* < 0.0001, and r = −0.372, *p* = 0.005, respectively) but positively correlated with the fall prediction model score value (r = 0.527, *p* < 0.0001). Further mediation analysis shows that the UPDRS total score values serve as mediators between lower stride length, straight walking, and turning speed and higher fall prediction model score values. Our results highlighted the relationship among stride length and speed, clinical disease severity, and risk of falling. As decreased stride length and speed are hallmarks of falls, monitoring the changes of quantitative biomechanical measures along with the use of wearable technology in a longitudinal study can provide a scientific basis for pharmacology, rehabilitation programs, and selecting high-risk candidates for surgical treatment to reduce future fall risk.

## 1. Introduction

Falls in people with idiopathic Parkinson’s disease (PD) have multifactorial causation [1], but gait disturbance is a major contributor. It can present from the early stages of the disease [2], and increased fall risk is the most debilitating symptom in patients with PD. The prevalence of falls and recurrent falls increases with disease progression, though this trend may reverse in the final stages of PD as the patient becomes immobile. Therefore, investigating the risk factors of falls as well as the potential strategies for the prevention of future falls is of great importance. Several studies have proposed several risk factors for falls in PD, including previous falls, older age, disease duration, poor balance, advanced disease, cognitive function impairment, disease severity, presence of freezing of gait (FOG), more severe axial symptoms, dyskinesia, and reduced leg strength [3,4,5,6]. 

Clinical approaches to quantifying the relationship between gait and falls in PD can broadly be divided into two: using clinical rating scales and quantitative biomechanical measures [7]. Several clinical scoring systems have been developed to assess the risk of falling [8,9]. The three-step fall prediction test is a valid and reliable scale for clinical fall risk prediction tools in PD [10]. However, utilizing only clinical scores to assess the risk of falling has some limitations. For example, the adequacy of these scales in terms of reliability and validity, and the specific mechanics associated with falls, has never been demonstrated. The assessment of clinical scales for PD is mostly completed through brief observations made during patient visits, and it can be subjective and affected by the examiner’s experience. It is also difficult to track changes in the motor function of patients between visits. Physicians are not able to make the most informed decisions relating to therapeutic strategies without frequent visits. In contrast, frequent clinic visits increase the physical and economic burden of patients and their families [11]. 

Alternatively, quantitative biomechanical measures, including spatiotemporal parameters and kinematic variables, represent the components of walking performance during a walking gait cycle and provide an individualized fall risk and highlight specific modifiable gait characteristics that can be targeted by gait training to minimize the future risk of falling [12]. During research on gait characteristics and falls in PD, the outcome of falls included the number of falls or presence/absence of falls. Spatiotemporal variables were the most reported biomechanical measures, and several parameters could reflect speed (step or stride time, cadence), amplitude (step or stride length), regularity of movement (step length variability), and balance control (trunk motion). Among them, walking speed was the most frequent [13,14,15,16], followed by stride or step measures (length, time, and time variability or asymmetry) [13,14,15,16,17], walking speed variability [14], and kinematic measures on head and trunk control [15,16], while kinematic measures on low extremity joint range of motion (ROM) (hip and knee) [16] and turning (speed, time, and step length) [18] were rarely reported in these previous articles. 

According to our hypotheses, clinical disease severity may contribute to reduced stride length and speed during the walking cycle, and clinical disease severity might also lead to an increased risk of falling in patients with PD. Thus, this raises the possibility that clinical disease severity might mediate the relationship between stride length and speed and the risk of falling in patients with PD. Our results may be beneficial for better awareness of the risk of falling in patients with PD and guide interventions to improve their long-term quality of life.

## 2. Patients and Methods

### 2.1. Study Design and Patient Selection

This single-center case-control study was conducted at a tertiary medical center and the main referral hospital in southern Taiwan. We evaluated 100 patients with a definitive diagnosis of idiopathic PD according to clinical diagnostic criteria [19]. The inclusion criteria were as follows: (1) patients who were followed up for more than six months after titration of their daily anti-Parkinsonian agents to a steady dose at the Neurology Outpatient Clinics and (2) patients who had Hoehn and Yahr stage 1–3 and could walk independently [20]. The exclusion criteria were (1) newly diagnosed PD or follow-up for less than six months; (2) neurological signs not related to the diagnostic criteria of PD; (3) advanced PD stage (Hoehn and Yahr staging equal or more than four) and inability to walk independently, including those who needed an assistive device during their assessments; (4) mild to moderate dementia (Clinical Dementia Rating (CDR) more than or equal to 1), which precluded following our instructions; and (5) any etiologies that were severe enough to interfere with balance, including visual problems and motor weakness caused by disuse of the lower limbs, spinal stenosis with radiculopathy, knee osteoarthritis, and any etiologies of polyneuropathy. The study was approved by the Institutional Review Committees on Human Research of the hospital (IRB 201901802B0). All participants received verbal and written information about the purpose and methodology of our research, and they signed an informed consent. A total of 39 patients were excluded from this study. Twenty-five out of the thirty-nine cases had advanced PD stage, seven had moderate dementia, four had spinal stenosis with radiculopathy, and the remaining three had knee osteoarthritis. Finally, only 61 patients were included in the analysis. 

### 2.2. Severity and Subtypes of PD

Four patients experienced severe motor fluctuations and needed to take the last medication at midnight. The remaining 57 patients had taken the last medication at last dinner the previous day and maintained an off state of medication until the next morning for examination. The clinical assessments, including the clinical rating scales and gait analyses, were performed during the “off” state of medication, which was defined as 12 h after the latest dose of anti-parkinsonism agents, and the “on” state, which was at least one hour after taking anti-parkinsonism agents. Complete medical history was recorded, including age at disease onset, sex, height and weight, body mass index (BMI), disease duration, and levodopa equivalent dose (LED) [21]. The clinical severity of PD was assessed using the Unified Parkinson’s Disease Rating Scale (UPDRS) and Hoehn and Yahr stages [20]. The UPDRS total score was computed as the sum of the UPDRS subscores I, II, and III. Furthermore, the UPDRS-derived postural instability and gait difficulty (PIGD) score, which ranges from 0 to 20, with higher scores reflecting greater PIGD severity, were also assessed [22,23]. The presence of freezing of gait (FOG) was assessed using the New Freezing of Gait Questionnaire (NFOG-Q) [24]. One neurologist (C.Y.L.) evaluated the UPDRS scores, H-Y staging, and NFOG-Q blind to the patients. The Cognitive Abilities Screening Instrument (CASI C-2.0), which consists of 20 items divided into nine domains, and the sum of the scores ranges from 0 to 100, with higher scores indicating better cognitive ability, were also calculated [25]. 

### 2.3. Severity and Frequency and Situations of Falling

The severity of falling within the last 6 months before enrollment were recorded. The severity of a fall was either mild (did not need any treatment), moderate (needed simple bandages treatment), or severe (needed to visit a hospital) [4]. A faller was defined to have at least one fall either indoors or outdoors during the preceding three months before enrollment was recorded [26]. Furthermore, we used the three-step clinical prediction tool to assess the risk of falling within the next 6 months in patients with PD [10]. It consists of three parts, including the history of falls in the past 12 months (yes = 6 points, no = 0), history of freezing of gait (yes = 3 points, no = 0), and self-selected comfortable gait speed < 1.1 m per second (yes = 2 points, no = 0). The total score ranged from 0 to 11 points by summing the points obtained for each of the three tests, and the probability of falling was as follows: low risk (score = 0), moderate risk (score = 2–6), and high risk (score = 8–11). 

### 2.4. Assessment of Gait Analysis

Three-dimensional Kinect V2 detectors were used to automatically track skeletal data and rebuild 25 significant reference joint points, using a sample rate of 30 Hz, Windows 10 operating system, i5 CPU or higher, and a specialized algorithm (GaitBEST, LONGGOOD MEDITECH LTD., Taipei, Taiwan). This gait analysis system automatically positioned the human body using Kinect detectors, capturing the location of joint points and fitting them into the main software program to further analyze and calculate key time points and displacement values. The results can be displayed immediately after every measurement. Except for poor reliability in ankle detection [27], the clinical study has shown good validity of kinematics for Kinect-based systems [28]. The spatiotemporal and kinematic parameters during the walking gait cycle were assessed after verbal instructions to initiate gait with the most involved leg for PD patients, walk 4–4.5 m, turn 180° after crossing a line on the ground, and return to the initial starting position. 

During the initial acceleration phase, the system automatically removes the data of 0.3 m, and during the deceleration period, it removes the data of the last 1.0 m to minimize the impact of the acceleration and deceleration period. We use several methods to overcome the presence of FOG during the walk cycle (e.g., carry a laser pointer and shine the laser in front of the patient’s foot). However, if the presence of FOG significantly exceeds the normal testing time, the system will automatically stop, and thus, it is excluded from the analysis and the same test needs to be repeated. The researcher can observe the visualized variation caused by FOG on the output report, which shows the ROM plot during the whole walking period. Three successful trials were analyzed and averaged during the off and on phases. The variables for spatiotemporal and low-extremity kinematic parameters were obtained for each participant. For each patient, the mean ± standard deviation (SD) of each variable for the three successful trials was calculated and averaged. The coefficient of variation (CV) of step length and time was calculated using all measurements obtained during the walk cycles and it was considered as a measure of the variability of the step length and time, respectively.

### 2.5. Assessment of the Functional Mobility

Optimal walking speed (OWS) is the walking speed adjusted for the lower limb length or height effects (OWS = sqrt (0.25 × 9.81 × lower limb length (or 0.54 of height)) [29]. In this situation, the speed is optimized and the metabolic energy expenditure is minimized. Locomotor rehabilitation index (LRI) [30], which expresses the relationship between the walking speed and OWS, is a complementary analysis of spatiotemporal parameters in gait assessment (LRI (%) = 100 × walking speed/OWS). We calculated OWS and LRI between the faller and non-faller groups in this study. 

### 2.6. Statistical Analysis 

In this study, the spatiotemporal and kinematic variables between fallers and non-fallers were compared by using an independent t-test. To reduce the chance of a Type I error (false positive), we employed the Bonferroni correction to each test of spatiotemporal and kinematic variables during the walking gait cycle. If we conduct 26 comparisons, we only reject the null hypothesis of each comparison if it has a *p*-value less than 0.002. We also employ the Receiver operating characteristic (ROC) curves were generated for the significant parameters between fallers and non-fallers for predicting the presence of falls (Figure 1). Only the *p*-value < 0.05 were listed in Figure 1. The spatiotemporal and kinematic variables before and after dopamine replacement therapy were compared using the paired *t*-test. The repeated-measures ANOVA was used to compare the parameter of LRI before and after dopamine replacement therapy between fallers and non-fallers during the straight walking and turning of the walking cycle, with adjusting sex and age as potential confounding variables. Correlation analysis was used to evaluate the relationship between the three-step fall prediction model score and UPDRS total score, and the parameters of spatiotemporal and kinematic parameters and clinical rating scores. Multiple linear regression analysis, using a stepwise procedure, was performed to evaluate the influence of independent variables on the mean score of the three-step fall prediction model (dependent variable). Finally, mediation analysis was used to test whether the direct effect of stride length (Figure 2A), straight walking (Figure 2B), and turning speed (Figure 2C) (independent variables) on risk of falling (dependent variable) can be explained by the indirect influence of the UPDRS total score (mediator) (Figure 2A–C). The statistical significance threshold was set at 0.05 in the Sobel test for all the relevant paths [31]. All statistical analyses were conducted using the IBM SPSS Statistics v23 software (IBM, Redmond, WA, USA). 

## 3. Results

### 3.1. General Characteristics of Patients with PD

The 61 patients with PD in this study included 32 women and 29 men. The baseline patient characteristics including age, sex, height (m), body weight (kg), body mass index (kg/m^2^), waist circumference (cm), disease duration, LED (mg), UPDRS I, II, III, total score, and UPDRS-derived PIGD score, Hoehn and Yahr stages, presence of FOG, dyskinesia, motor fluctuation, “off” dystonia, and cognitive abilities screening instrument are presented in Table 1. Furthermore, 12 out of the 23 cases who had FOG by history had experienced FOG during the walking gait cycle.

### 3.2. Baseline Spatiotemporal and Kinematic Variables 

The parameters of the spatiotemporal and kinematic variables during the walking gait cycle between fallers and non-fallers are listed in Table 2. Straight speed (m/s), stride length (m), step length (m), and turning speed were significantly lower (*p* = 0.002, *p* < 0.0001, *p* < 0.0001, and *p* = 0.002, respectively), while step time variability (CV) was higher (*p* = 0.002) during the off phase in the faller group. The other spatiotemporal and kinematic variables in the off phase were not statistically significant between the two groups. Furthermore, the spatiotemporal and kinematic variables in the on phase were only stride length (m) and step length (m) (*p* = 0.03 and *p* = 0.03), which showed statistical significance between the two groups (Table 2). The Bonferroni-adjusted *p*-values showed only stride length, step length, straight forward speed, and turning speed had statistical significance (*p* < 0.002) or marginal statistical significance. Furthermore, the significant spatiotemporal and kinematic variables in the presence of falls according to the independent t-test, as listed in Table 2, were tested by using an ROC curve analysis and showed that only stride length (AUC = 0.728, *p* = 0.007), step length (AUC = 0.728, *p* = 0.007), straight forward speed (AUC = 0.687, *p* = 0.028), and turning speed (AUC = 0.709, *p* = 0.014) had diagnostic accuracy in the presence of falls (*p* < 0.05) (Figure 1). 

### 3.3. Effect of Dopamine Replacement Therapy on Spatiotemporal and Kinematic Variables 

The spatiotemporal and kinematic variables during the walking gait cycle before and after dopamine replacement therapy are listed in Table 3. Speed (m/s), stride length (m), and step length (m) were significantly increased after dopamine replacement therapy (*p* = 0.004, *p* < 0.0001, and *p* = 0.001, respectively). Total hip ROM (degrees) significantly increased after dopamine replacement therapy (*p* < 0.0001). Step length variability (CV) and turning time (s) decreased after dopamine replacement therapy (*p =* 0.01 and *p* = 0.01, respectively). 

### 3.4. Effect of Dopamine Replacement Therapy on Functional Mobility

The comparison of LRI between non-fallers and fallers before and after dopaminergic therapy was listed in Table 4 and Figure 3. The LRI values on straight waking and turning walking in the off phase were lower among the fallers (*p* = 0.002 and *p* = 0.02, respectively). The LRI values on straight waking and turning walking in the on phase were similar but slightly higher among the non-fallers (*p* = 0.51 and *p* = 0.14, respectively). 

The LRI values on straight waking in the fallers showed increased after dopaminergic therapy (*p* = 0.001), but the LRI values on turning among the fallers did not show statistical significance after dopaminergic therapy (*p* = 0.23). The LRI values of straight walking and turning among the non-fallers did not show statistical significance after dopaminergic therapy (*p* = 0.30 and *p* = 0.70). Furthermore, the LRI values on straight walking between fallers and non-fallers before and after dopaminergic therapy by means of a repeated measure ANOVA after controlling age and sex showed statistical significance (*p* = 0.027). 

### 3.5. Correlation Analyses of Spatiotemporal and Kinematic Parameters, as well as Baseline Characteristics on the Three-Step Fall Prediction Model Score

The correlation analyses were used to test the influence of the parameters of spatiotemporal and kinematic parameters and baseline characteristics on the three-step fall prediction model score and UDPRS total score. The statistical results (correlation coefficient, *p*-value) were listed in Table 5. Speed (m/s), stride length (m), step length (m), turning speed, turning step length, and hip range of motion (ROM) (°) and knee ROM (°) were negatively correlated with both three-step fall prediction model score and UDPRS total score, while disease duration (years), step length variability (CV), and step time variability (CV) were positively correlated with both the three-step fall prediction model score and UDPRS total score. 

### 3.6. Clinical Factors Significantly Associated with the Three-Step Fall Prediction Model Score 

The effects of the variables for the three-step fall prediction model are listed in Table 6. We used only the significant variables in Table 5 for the multiple linear regression analysis model to identify the crucial factors that influence the augmented three-step fall prediction model score in patients with PD. The results revealed that UPDRS total score and stride length (m) during the ‘off’ stage was significantly associated with the three-step fall prediction model score (Table 6). 

### 3.7. Mediation Analysis for the Severity of Spatiotemporal Parameters, Clinical Disease Severity, and Risk of Falling 

The primary hypothesis of this analysis concerns whether the effect of stride length and speed during straight walking and turning (independent variable) on the risk of falling (three-step fall prediction model score, dependent variable) was explained indirectly by UPDRS total scores (mediator) with a significant group main effect. The path model jointly tested three effects of interest that are required if clinical disease severity (UPDRS total score) links the severity of spatiotemporal parameters (stride length and speed) with the risk of fallings (three-step fall prediction model score): (a) the effect of the independent variable (severity of spatiotemporal parameters including stride length, straight walking, and turning speed value) on the mediator (UPDRS total score value; indirect effect, path a); (b) the effect of the mediator on the dependent variable (three-step fall prediction model score) by controlling the effect for the UPDRS total score value (indirect effect, path b); and (c) the mediation effect a × b, which is defined as the reduction of the relationship between the independent and dependent variables (severity of spatiotemporal parameters and three-step fall prediction model score value; total relationship, path c) by including the mediator into the model (direct path, path c′). For simplicity, we report a full list of the results from the present study that fulfill the three criteria cited previously. All the mediation relationship was significant (*p* = 0.018, partial mediation, *p* = 0.003, partial mediation and *p* = 0.017, partial mediation, respectively, Sobel test) (Figure 2A–C) (Table 7).

## 4. Discussion

### 4.1. Major Findings of Our Study

Consistent with our hypothesis and in line with the extant literature, PD patients who had fallen experienced lower stride length and speed, higher UPDRS total score, and higher three-step falls prediction model score than those patients who had not. Thus, our results highlighted the relationship among stride length and speed, clinical disease severity, and risk of falling. At present, efforts to reduce future risks of falling in patients with PD remain unsatisfactory. Monitoring the worsening or improvement of quantitative biomechanical measures during follow-up studies or the evaluation of therapeutic regimens is important for improving the quality of life of patients; however, it is a challenge for clinicians. 

### 4.2. Spatiotemporal and Kinematic Parameters Associated with Risk of Falling

PD subjects have difficulty with the internal regulation of stride length, even though cadence control is intact. The simple equation for speed is equal to stride length × cadence. If we increase either stride length or cadence, then we will go faster. Otherwise, if only stride length is decreased but cadence remain intact, the speed is also decreased. As the disease progresses, subjects have a higher cadence rate than control subjects as compensation for reduced stride and step length [32]. Furthermore, many studies, including a recent meta-analysis, have found cadence to be increased in PD [33]. Our study enrolled those patients who had Hoehn and Yahr stage 1–3 and could walk independently and showed that those who had fallen appeared to have a decreased stride, step length, and speed, but cadence did not show the significant difference as compared with those who were non-fallers. 

### 4.3. The Effects of Dopamine Replacement Therapy on the Walking Gait Cycle

Levodopa has been shown to improve bradykinetic and hypometric spatial characteristics of gait (speed and step length) by increasing ability to increase step length upon ambulation speed increase [34]. Our study demonstrated that speed (m/s), stride and step length (m), step length variability, and total hip ROM significantly increased after dopamine replacement therapy. Dopamine replacement therapy may improve speed and stride length, as well as decrease step length variability corresponding to increased hip ROM [35], ultimately making gait more efficient. Although levodopa has been shown to improve bradykinesia and rigidity, treatment with levodopa increases postural sway abnormalities in patients with PD [36]. It has also shown a limited response to levodopa replacement therapy during the late stages of PD [37]. Motor complications, including dyskinesia, motor fluctuation, and off dystonia, could occur in advanced PD. Once dopaminergic therapy has been started, off dystonia may appear when there is a decrease in brain dopamine levels, while levodopa-induced dyskinesia may compromise balance and contribute to postural instability and falls. Our patient who had dyskinesia also had a higher risk of falls. Some research has hypothesized that decreased variability may actually be a negative sign, as the complexity of a system allows for a better response to unexpected perturbations. This is put forward as one reason that, while levodopa improves many disease factors, it does not always positively impact PIGD and falls and may worsen them. Turning during the walking cycle is often difficult because it requires a series of gait initiations. One-third of our patients had FOG and it often occurs when turning [38]. It also has a limited response to dopaminergic drugs and is episodic and unpredictable [38]; moreover, it may explain why dopaminergic therapy did not cause statistically significant improvement of the speed and step length during turning movements.

### 4.4. Three Candidate Variables in the Mediation Analysis Model

In this study, we propose the hypothesis that clinical disease severity might mediate the relationship between severity of spatiotemporal parameters and the risk of falling in PD. Clinical studies have demonstrated that in patients with mild PD without significant cognitive impairment, falls can be predicted with a high degree of accuracy using the simple three-test clinical tool [10]. However, the severity associated with the score based on this model and its correlations with the quantitative biomechanical measures of gait has not been evaluated. Our study demonstrated that the three-step fall prediction model score (score 0–11) was significantly correlated with disease severity (disease duration and LED) and the biomechanical measures of gait (e.g., stride length (m), straight forward and turning speed (m/s), step length (m), step length variability (CV), turning time (s), and total hip and knee ROM (degree)). 

The UPDRS-derived PIGD score [39], which comprised UPDRS items 13, 14, 15, 29, and 30, fulfills the criteria for a “recommended” scale and has adequate clinometric characteristics. However, only a few aspects of gait and balance are addressed and floor effects make the score is less suitable as an outcome measure in patients with mild disease [39]. Our study enrolled those patients who could walk independently and eight out of 61 cases whose UPDRS-derived PIGD score had floor effects (the value of the PIGD score is equal to zero). Thus, we finally chose UDPRS total score as reflective of clinical disease severity. 

Regarding the spatiotemporal parameters, decreased stride length during walking is a hallmark of PD, as shown in our previous study [32], and the change in speed is in proportion to the change in stride length. A self-report was used to assess the presence of falls during the preceding three months before enrollment is reliable [26] and we compared continuous variables of gait parameters between the fallers and non-fallers by means of independent t-tests. To reduce the chance of a Type I error (false positive), we further employed both the Bonferroni correction and AUC under the ROC curve for the presence of falls of each significant gait parameter associated with fallers in the independent t-test (Table 2). Only stride length, step length, straight forward speed, and turning speed had the diagnostic accuracy in the presence of fallers (*p* < 0.05 in ROC curve (Figure 1) and *p* < 0.002 in Bonferroni correction). Stride length is approximately double the step length. Thus, we finally chose those parameters with diagnostic accuracy for the fallers (*p* < 0.05) in ROC analysis, including stride length and straight forward and turning speed at an off phase, as the severity of spatiotemporal parameters. Difficulty turning is a well-known contributor to mobility disability and risk of falls in older people [40]. One study demonstrated that quality of turning, including mean turn duration, mean peak speed, and mean number of steps to complete a turn, was lower in recurrent fallers compared to non-fallers and single-fallers [40]. Our study demonstrated that turning speed during the ‘off-stage rather than the ‘on’ stage was significantly lower in those patients who had retrospective falls. In contrast to LRI in straight walking, LRI in turning walking did not show a significant improvement after dopaminergic therapy. Our study also highlighted the hypothesis that clinical disease severity mediates the relationship between turning speed and the risk of falling in patients with PD.

## 5. Strengths and Limitations 

This study has several strengths. First, our study confirmed that decreased stride f straight and turning speed are hallmarks of falls. Although dopamine replacement therapy may improve speed and stride length, gait progression was mostly unrelated to dopaminergic medication adjustments over 6 years of follow-up in one study [7]. This highlights the limitations of current dopaminergic therapy and the need to improve interventions targeting gait decline. Second, our results highlight the relationship among stride length and speed, clinical disease severity, and risk of falling. In the past, clinical disease severity was often viewed as a confounding factor for gait research in PD. In contrast, the overall objective of mediation analysis is to make causal inferences about mechanisms. It can not only investigate causal paths between gait parameters and the fall risk but also can examine the extent to which clinical disease severity explains the effect of exposure on fall risk. Third, evidence suggests that disease severity plays a critical role in postural instability, which increases the risk of falling [6,37,41]. Higher dopaminergic dosing could induce dyskinesia, which may compromise balance and contribute to postural instability and falls in advanced PD [36]. However, surgery (e.g., deep brain stimulation) can not only correct abnormal sensory organizational components but also improve the motor components of postural control [36,41,42]. The quantitative biomechanical measures can provide a scientific basis to track disease progression and select high-risk candidates for surgical treatment to minimize the future risk of falling.

Our study has several limitations. First, it is well known that increased fall risk is the most debilitating symptom in patients with PD. However, we excluded those who had advanced PD and could not walk independently because of safety issues and limitations of gait analysis devices. Second, the study enrolled those with low extremity kinematics (hip and knee) during the walking gait cycle, and the kinematics of the head, trunk, and shoulder were not analyzed. Therefore, we did not analyze the kinematic measures of the head, trunk, and shoulder control on the risk of falling in patients PD. Third, gait analysis system setting (e.g., acceleration and deceleration phases) and patient’s conditions (e.g., presence of FOG during the walking cycle) could also influence the value of parameters in the walking cycle and exist a bias. Fourth, the ability of our quantitative biomechanical measures to track disease progression and predict the risk of falling in a longitudinal follow-up study is unclear, and further longitudinal follow-up is mandatory. Although the sample size is not large, the numbers of variables considered for the multiple linear regression analysis is also not large. Furthermore, based on the stepwise procedures and collinearity statistics, only two variables were selected as the most important variables predicting the falling risks. Thus, the maximum likelihood estimates of the coefficients are valid in the analysis. Finally, several clinical factors (e.g., clinical disease severity, dopaminergic medication dosage, etc.) have often been viewed as confounding factors for gait research in PD. More sophisticated mediation analysis models (e.g., moderated mediation with two independent moderators or two interacting moderators) can provide a better understanding of the mechanism of falls in patients with PD in future research.

## 6. Conclusions

As decreased stride length and speed are hallmarks of falls, monitoring the changes of quantitative biomechanical measures along with the use of wearable technology in a longitudinal study can provide a scientific basis for pharmacology, rehabilitation programs, and selecting high-risk candidates for surgical treatment to reduce future fall risk.

## Figures and Tables

**Figure 1 jpm-12-00192-f001:**
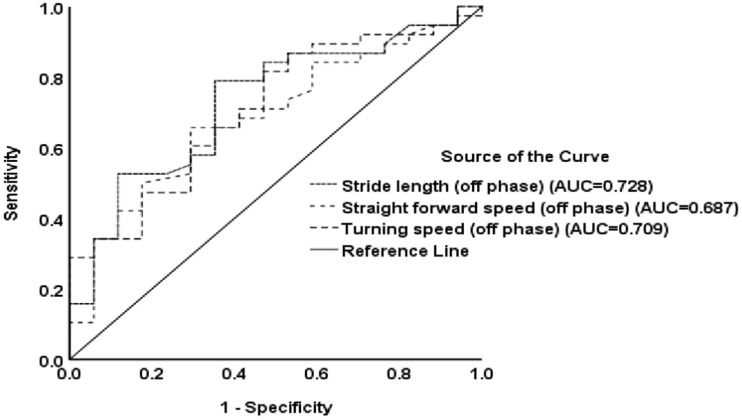
Receiver operator characteristic curves for predicting the presence of falling. The diagnostic accuracy of the spatiotemporal parameters is shown based on the receiver operating characteristic curve analysis (*p* < 0.05).

**Figure 2 jpm-12-00192-f002:**
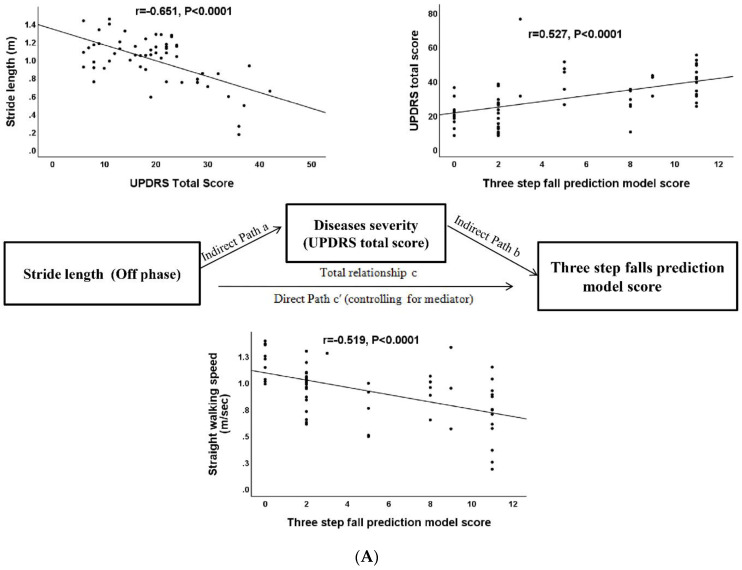
The diagram of the mediation hypothesis framework. The mediation analysis tests whether the direct effect of stride length (**A**), straight walking (**B**), and turning speed (**C**) (independent variables) on risk of falling (dependent variable) can be explained by the indirect influence of the UPDRS total score (mediator).

**Figure 3 jpm-12-00192-f003:**
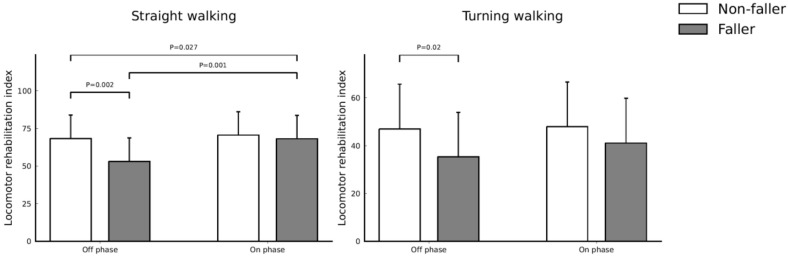
Comparison of the locomotor rehabilitation index between non-fallers and fallers before and after dopaminergic therapy.

**Table 1 jpm-12-00192-t001:** Baseline characteristics of Parkinson’s Disease.

	Non-Fallers (*n* = 39)	Fallers (*n* = 22)	*p*-Value
Age, years	67.9 ± 8.6	66.3 ± 8.9	0.48
Sex (men/women)	21/18	8/14	0.19
Height (m)	1.60 ± 0.08	1.57 ± 0.08	0.14
Body mass (kg)	64.8 ± 10.1	61.5 ± 12.6	0.26
Body mass index (kg/m^2^)	25.4 ± 3.8	25.1 ± 5.3	0.79
Waist circumference (cm)	89.4 ± 9.3	87.8 ± 11.3	0.56
Disease duration, years	4.7 ± 3.7	8.9 ± 5.3	0.001 *
Total LED (mg)	612.9 ± 348.5	1312.3 ± 745.4	<0.0001 **
UPDRS total score ^α^ (off phase)	25.3 ± 13.7	37.2 ± 10.5	0.001 *
UPDRS I ^β^ (off phase)	1.2 ± 1.1	2.1 ± 1.5	0.008 *
UPDRS II ^γ^ (off phase)	7.8 ± 4.6	14.5 ± 4.8	<0.0001 **
UPDRS III ^δ^ (off phase)	16.3 ± 10.0	20.3 ± 7.2	0.11
UPDRS-derived PIGD score (off phase) ^Ω^	2.8 ± 2.3	7.6 ± 3.6	<0.0001 **
Hoehn and Yahr stages	1.7 ± 0.8	2.5 ± 1.0	0.001 *
Freezing of gait	8	15	<0.0001 **
Dyskinesia	3	7	0.02 *
Motor fluctuation	4	9	0.004 *
“Off” dystonia	4	5	0.23
Cognitive Abilities Screening Instrument	82.2 ± 16.0	82.1 ± 13.9	0.99
Underlying diseases			
Hypertension	15	6	0.38
Hyperlipidemia	27	9	0.03
Coronary artery diseases	4	4	0.44
Falling severity ^Ф^			
Mild	0	6	
Moderate	0	12	
Severe	0	0	

* Indicates that *p*-value < 0.05; ** indicates that *p*-value < 0.01; Data are presented as mean +/− SD or number. Abbreviation: UPDRS = Unified Parkinson’s Disease Rating Scale; LED = Levodopa equivalent dose; ^α^ = Total score is the combined sum of the I, II, and III subscore. ^β^ = I. Mentation, behavior, and mood subscore. ^γ^ = II. Activities of daily living (ADL) subscore. ^δ^ = III. Motor subscore; ^Ω^ = The value in eight patients was zero; ^Ф^ = only analyzed those 22 patients who had recurrent falls.

**Table 2 jpm-12-00192-t002:** Baseline spatiotemporal and kinematic variables during the walking gait cycle.

	Non-Fallers (*n* = 39)	Fallers (*n* = 22)	*p*-Value
Spatiotemporal parameters (off phase)			
Straight forward			
Cadence (steps/s)	1.84 ± 0.17	1.85 ± 0.22	0.78
Speed (m/s)	1.0 ± 0.23	0.77 ± 0.29	0.002 *
Stride length (m)	1.08 ± 0.21	0.82 ± 0.28	<0.0001 **
Step length (m)	0.54 ± 0.10	0.41 ± 0.14	<0.0001 **
Step length variability (CV)	14.76 ± 5.0.2	21.08 ± 13.12	0.05
Stride Time (s)	1.09 ± 0.10	1.10 ± 0.15	0.72
Step Time (s)	0.55 ± 0.04	0.55 ± 0.08	0.91
Step Time variability (CV)	11.58 ± 3.76	18.76 ± 13.04	0.002 *
Turning			
Turning time (s)	2.45 ± 0.98	3.19 ± 1.59	0.09
Turning speed (m/s)	0.69 ± 0.28	0.51 ± 0.14	0.002 *
Turning step length (m)	0.42 ± 0.08	0.38 ± 0.07	0.09
Low extremities kinematics (off phase)			
Total hip ROM (°)	53.83 ± 9.28	48.53 ± 11.60	0.06
Total Knee ROM (°)	67.61 ± 6.55	65.08 ± 10.06	0.25
Spatiotemporal parameters (on phase)			
Straight forward			
Cadence (steps/s)	1.83 ± 0.18	1.93 ± 0.22	0.07
Speed (m/s)	1.03 ± 0.20	0.98 ± 0.21	0.40
Stride length (m)	1.11 ± 0.16	1.02 ± 0.17	0.03 *
Step length (m)	0.56 ± 0.08	0.51 ± 0.09	0.03 *
Step length variability (CV)	14.11 ± 4.78	15.03 ± 5.23	0.49
Stride Time (s)	1.10 ± 0.11	1.05 ± 0.14	0.15
Step Time (s)	0.55 ± 0.06	0.53 ± 0.07	0.14
Step Time variability (CV)	12.33 ± 3.49	13.61 ± 5.49	0.27
Turning			
Turning time (s)	2.24 ± 0.63	2.65 ± 1.22	0.15
Turning speed (m/s)	0.70 ± 0.23	0.59 ± 0.28	0.12
Turning step length (m)	0.44 ± 0.07	0.42 ± 0.12	0.44
Low extremities kinematics (on phase)			
Total hip ROM (°)	55.64 ± 7.53	56.02 ± 8.63	0.86
Total Knee ROM (°)	68.38 ± 6.09	67.67 ± 6.68	0.67

* Indicates that *p*-value < 0.05; ** indicates that *p*-value < 0.01. Abbreviations: CV = coefficient of variation; ROM (degree) = range of motion; degree = °. ^ξ^ = only *p* < 0.05 by independent *t*-tests were analyzed.

**Table 3 jpm-12-00192-t003:** Effect of dopamine replacement therapy on spatiotemporal and kinematic variables during the walking gait cycle.

	Off Phase	On Phase	*p*-Value	Normal Reference ^ф^
Spatiotemporal parameters				
Straight forward				
Cadence (steps/s)	1.83 ± 0.19	1.85 ± 0.20	0.49	1.82 ± 0.16
speed (m/s)	0.91 ± 0.28	1.0 ± 0.20	0.004 **	1.33 ± 0.32
Stride length (m)	0.99 ± 0.27	1.08 ± 0.18	<0.0001 **	1.17 ± 0.24
Step length (m)	0.50 ± 0.13	0.54 ± 0.09	0.001 **	0.58 ± 0.12
Step length variability (CV)	17.25 ± 9.35	14.44 ± 5.13	0.01 *	14.03 ± 14.56
Stride time (s)	1.10 ± 0.12	1.09 ± 0.13	0.85	0.90 ± 0.18
Step time (s)	0.55 ± 0.06	0.55 ± 0.06	0.55	0.47 ± 0.32
Step time variability (CV)	14.25 ± 9.1	13.05 ± 4.44	0.30	11.88 ± 8.69
Turning				
Turning time (s)	2.71 ± 1.27	2.27 ± 0.63	0.01 *	1.77 ± 0.40
Turning speed (m/s)	0.62 ± 0.25	0.66 ± 0.26	0.37	0.79 ± 0.09
Turning step length (m)	0.41 ± 0.08	0.43 ± 0.10	0.11	0.52 ± 0.04
Low extremities kinematics				
Total hip ROM (°)	52.17 ± 10.55	55.70 ± 8.29	<0.0001 **	66.52 ± 6.65
Total Knee ROM (°)	66.74 ± 7.94	68.15 ± 6.27	0.08	70.97 ± 3.31

* Indicates that *p*-value < 0.05; ** indicates that *p*-value <0.01; ^Ф^ = Unpublished data of normal reference are from 300 healthy volunteers. We only show the data but do not make a statistical comparison with the patient group. Abbreviations: LED, levodopa equivalent dose; CV = coefficient of variation; ROM (degree) = range of motion; degree = **°**.

**Table 4 jpm-12-00192-t004:** Comparison of locomotor rehabilitation index between non-fallers and fallers before and after dopaminergic therapy.

	Non-Fallers (*n* = 39)		Fallers (*n* = 22) ^β, γ^	
	Off Phase ^α^	On Phase	Off Phase ^α^	On Phase
Optimal walking speed (m/s)	1.45 ± 0.03		1.44 ± 0.0.3	
LRI (%) during straight walking	68.42 ± 15.63 #	70.68 ± 13.15	53.19 ± 20.03	68.27 ± 14.25 §
LRI (%) during turning walking	47.03 ± 18.67 #	47.97 ± 15.91	35.34 ± 10.57	41.13 ± 19.31

Values are expressed in mean ± SD unless otherwise indicated. Abbreviations: OWS = optimal walking speed; LRI = Locomotor rehabilitation index. ^α^ = LRI during off phase between non-faller and faller groups were compared by means of an independent *t*-test, # = Indicated *p* < 0.05. ^β^ = Before and after dopaminergic therapy in the faller group were compared by paired *t* test = Indicated *p* < 0.05. ^γ^ = LRI in faller and non-faller groups before and after dopaminergic therapy by means of repeated measure ANOVA, § = Indicated *p* < 0.05.

**Table 5 jpm-12-00192-t005:** Correlation analysis between the three-step fall prediction model score and UPDRS total score, and parameters of spatiotemporal and kinematic parameters.

Variables	Three-Step Fall Prediction Model Score		UPDRS Total Score	
	r	*p*-Value	r	*p*-Value
Baseline characteristics				
Age, years	−0.094	0.474	−0.047	0.724
Disease duration, years	0.520	<0.0001 **	0.441	<0.0001 **
UPDRS total score	0.527	<0.0001 **	---	---
Cognitive Abilities Screening Instrument	−0.125	0.425	−0.241	0.107
Levodopa equivalent dose	0.519	<0.0001 **	0.522	<0.0001 **
Spatiotemporal parameters (off phase)				
Cadence (steps/s)	0.005	0.97	−0.087	0.518
Speed(m/s)	−0.519	<0.0001 **	−0.555	<0.0001 **
Stride length(m)	−0.583	<0.0001 **	−0.651	<0.0001 **
Step length(m)	−0.582	<0.0001 **	−0.601	<0.0001 **
Step length variability (CV)	0.379	0.004 **	0.405	0.002 **
Stride Time (s)	0.089	0.51	0.133	0.321
Step Time (s)	0.047	0.731	0.152	0.156
Step Time variability (CV)	0.43	0.001 **	0.465	<0.0001 **
Double support time (s)	0.26	0.051	0.361	0.005
Single support time (s)	−0.043	0.748	0.153	0.252
Swing phase (s)	−0.044	0.744	0.129	0.334
Stance phase (s)	0.123	0.363	0.182	0.171
Turning time (s)	0.394	0.003 **	0.501	<0.0001 **
Turning speed (m/s)	−0.462	<0.0001 **	−0.372	0.005 **
Turning step length (m)	−0.352	0.0009 **	−2.93	0.03 *
Low extremities kinematics (off phase)				
Total hip ROM (°)	−0.376	0.0004 **	−0.512	<0.0001 **
Total Knee ROM (°)	−0.27	0.043 *	−0.460	<0.0001 **

r: correlation coefficient. * Indicates that *p*-value < 0.05; ** indicates that *p*-value < 0.01. Abbreviations: LED, levodopa equivalent dose; CV = coefficient of variation; ROM (degree) = range of motion; degree = °.

**Table 6 jpm-12-00192-t006:** Effects of the variables on the three-step fall prediction model score in multiple linear regression according to a correlation analysis.

Significant Univariable	Three-Step Fall Prediction Model Score	
	Regression Coefficients	*p*-Value
Constant	6.58	0.056
UPDRS total score	0.14	0.003 *
Stride length (m)	−5.62	0.025 *

r: correlation coefficient. * Indicates that *p*-value < 0.05. Abbreviations: UPDRS, Unified Parkinson’s Disease Rating Scale; Model summary: Model: Predictors: (constant), UPDRS total score, Stride length (m); r^2^ = 0.395.

**Table 7 jpm-12-00192-t007:** Potential UPDRS total score mediators of stride length and speed on the risk of falls.

Mediation Analysis model ^Ω^	Path Coefficient	Standard Error	*p*-Value	Sobel Test
Model 1: X = stride length				
Total effects, path c	−9.22	1.73	<0.0001	
Direct effects, path c′	−5.26	2.02	0.012	0.018
Indirect effect, path a	−28.38	5.08	<0.0001	
Indirect effect, path b	0.16	0.04	<0.0001	
Model 2: X = Straight forward speed				
Total effects, path c	−7.81	1.74	<0.0001	
Direct effects, path c′	−3.90	1.87	0.04	0.003
Indirect effect, path a	−24.33	5.09	<0.0001	
Indirect effect, path b	0.16	0.035	<0.0001	
Model 3: X = Turning speed				
Total effects, path c	−7.32	1.95	<0.0001	
Direct effects, path c′	−4.34	1.90	0.027	0.017
Indirect effect, path a	−17.78	5.95	0.004	
Indirect effect, path b	0.16	0.035	<0.0001	

Mediation model: X = independent variable; fall prediction model score = dependent variable; UPDRS total score = mediator, Total effects, path c: The relationship between the independent and dependent variables. Direct effects, path c′: The relationship between the independent and dependent variables by including the mediator into the model. Indirect effect, path a: The effect of the independent variable on the mediator. Indirect effect, path b: The effect of the mediator on the dependent variable by controlling the effect for the independent variables. Ω = The mediation effects a × b, which is defined as the reduction of the relationship between the independent and dependent variables (total relationship, path c) by including the mediator into the model (direct path, path c′.

## Data Availability

The datasets used and/or analyzed during the current study are available from the corresponding author upon reasonable request.

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
