# Peer review of "Clinical Disease Severity Mediates the Relationship between Stride Length and Speed and the Risk of Falling in Parkinson’s Disease"

_jpm, 2022, doi:10.3390/jpm12020192_

Round 1

Reviewer 1 Report

The authors aim to demonstrate an association with PD disease severity and gait parameters particularly stride length, speed and risk of falling. This article adds to the literature of using objective biomechanical measurements may assist with determining the risk of falls and disease severity.

It is well known that changes to gait biomechanics are observed as PD progresses and rather some studies have revealed this may be present in earlier disease as well.  In this case control study, individuals were  assess both off and on dopaminergic therapy. Off medication was defined as 8-12 hours, however authors state “Except for a minority of patients who experienced motor fluctuations, most patients had taken the last medication…”. This statement is unclear, what do they mean by stating that there were some patients with motor fluctuations?

The severity of a fall is defined into mild, moderate, and severe based on the need for treatment, however what is the meaning of “simple treatment” and it is unclear if the frequency of falls contributes to the severity.

The authors appropriately note limitations of their study including the exclusion of individuals with advanced PD, and the presence of FOG and its impact on the gait, however I suggest the authors mention the impact of dystonia and on and/or off dose dyskinesias on gait. 

Did individuals use an assistive devices during their assessments if needed? This is not mentioned. 

Statistiacal analysis is appropriate.

This study informs future research into the field of using quatnitivate biomechanic measurements to possibly assist in disease progression and long-term studies are needed.

Figures 2a, b, c can be made more clear, appear busy with many graphs in each.

References should include more recent literature including: ie: Wilson et al, Gait Progression Over 6 Years in Parkinson’s Disease: Effects of Age, Medication, and Pathology Frontiers 2020,  

Conclusion should be strengthened to include that the quantitative biomechanics measured and its relation to disease severity, fall frequency and what future studies should consider assessing and types of interventions that can be implemented to improve these measures.  

Consider discussing future directions to include assessing gait biomechanics in DBS among individuals in various stages of disease, along with the use of wearable technology as a method to detect changes

Author Response

Comments and Suggestions for Authors

  1. The authors aim to demonstrate an association with PD disease severity and gait parameters particularly stride length, speed and risk of falling. This article adds to the literature of using objective biomechanical measurements may assist with determining the risk of falls and disease severity. It is well known that changes to gait biomechanics are observed as PD progresses and rather some studies have revealed this may be present in earlier disease as well. In this case control study, individuals were assessing both off and on dopaminergic therapy. Off medication was defined as 8-12 hours, however authors state “Except for a minority of patients who experienced motor fluctuations, most patients had taken the last medication…”. This statement is unclear, what do they mean by stating that there were some patients with motor fluctuations?

Answers: Thanks for your comment. There are 4 patients who experienced motor fluctuations and taken the last medication in the midnight. To make it clear, we revised the sentence into “Four patients experienced severe motor fluctuations and need to take the last medication at midnight. The remaining 57 patients had taken the last medication at last dinner the previous day and maintained an off state of medication until the next morning for examination”

  1. The severity of a fall is defined into mild, moderate, and severe based on the need for treatment, however what is the meaning of “simple treatment” and it is unclear if the frequency of falls contributes to the severity.

Answers: The severity of a fall is defined as mild, moderate, and severe based on the need for treatment in every episode, and the meaning of “simple treatment” indicates “simple bandage treatment”.  The frequency of falls did not contribute to the severity of our study. In the Patients and Method section, we change “simple treatment” into “simple bandage treatment” to make it clear.

  1. The authors appropriately note limitations of their study including the exclusion of individuals with advanced PD, and the presence of FOG and its impact on the gait, however I suggest the authors mention the impact of dystonia and on and/or off dose dyskinesias on gait.

Answers: Thanks for your comments. We added the following sentences in Discussion section according to your comment. They are as follows:

Motor complications including dyskinesia, motor fluctuation, and off dystonia could occur in advanced PD. Once dopaminergic therapy has been started, off dystonia may appear when there is a decrease in brain dopamine levels while levodopa-induced dyskinesia may compromise balance and contribute to postural instability and falls. Our patient who had dyskinesia also had a higher risk of falls.

  1. Did individuals use an assistive device during their assessments if needed? This is not mentioned.

Answers: Thanks for your comment. Those patients who need an assistive device during their assessments were exclude from our study. To make it clear, we added the sentence in Patients and Method section, and they are as follows:

The exclusion criteria were (1) newly diagnosed PD or follow-up for less than six months; (2) neurological signs not related to the diagnostic criteria of PD; (3) advanced PD stage (Hoehn and Yahr staging equal or more than four) and inability to walk independently who need an assistive device during their assessments

  1. Statistical analysis is appropriate.

Answers: Thanks for your comment.

  1. This study informs future research into the field of using quantitate biomechanics measurements to possibly assist in disease progression and long-term studies are needed.

Answers: Thanks for your comment.

  1. Figures 2a, b, c can be made clearer, appear busy with many graphs in each.

Answers: Thanks for your comment. Figures 2a, b, c was made clearer according to your comment.

  1. References should include more recent literature including: ie: Wilson et al, Gait Progression Over 6 Years in Parkinson’s Disease: Effects of Age, Medication, and Pathology Frontiers 2020,

Answers: We cited the reference you provide in our manuscript according to your comment.

  1. Conclusion should be strengthened to include that the quantitative biomechanics measured and its relation to disease severity, fall frequency and what future studies should consider assessing and types of interventions that can be implemented to improve these measures.

Answers: Thanks for your comment. We added the strength of our study to make it clear according to your comment. They are as follows:

This study has some strengths. First, our study confirmed that decreased stride length, and straight and turning speed are hallmarks of falls.  Although dopamine replacement therapy may improve speed and stride length, gait progression was mostly unrelated to dopaminergic medication adjustments over 6-years follow-up in one study. It highlights the limitations of current dopaminergic therapy and the need to improve interventions targeting gait decline. Second, our results highlight the relationship among stride length and speed, clinical disease severity, and risk of falling. In the past, clinical disease severity often has been viewed as a confounding factor for gait research in PD. In contrast, the overall objective of mediation analysis is to make causal inferences about mechanisms. It can not only investigate causal paths between gait parameters and the fall risk but also can examine the extent to which clinical disease severity explains the effect of exposure on fall risk. Third, evidence suggests that disease severity plays a critical role in postural instability, which increases the risk of falling. Higher dopaminergic dosing could induce dyskinesia may compromise balance and contribute to postural instability and falls in advanced PD. However, surgery (e.g., deep brain stimulation) not only can correct abnormal sensory organizational components but also improve the motor components of postural control. The quantitative biomechanical measures can provide a scientific basis to track disease progression and select high-risk candidates for surgical treatment to minimize the future risk of falling

  1. Consider discussing future directions to include assessing gait biomechanics in DBS among individuals in various stages of disease, along with the use of wearable technology as a method to detect changes

Answers: Thanks for your comment. We revised the conclusion according to your comment. They are as follows:

As decreased stride length and speed are hallmarks of falls, monitoring the changes of quantitative biomechanical measures along with the use of wearable technology in a longitudinal study can provide a scientific basis for pharmacology, rehabilitation programs, and select high-risk candidates for surgical treatment to reduce future fall risk.

Reviewer 2 Report

Main comments

Dear authors, the topic of functional mobility and gait characteristics in people with lumbar canal stenosis is timely and interesting. The article aimed to compare gait kinematics between fallers and non-fallers, and with the effect of medication. In addition, regression models were performed in order to understand the determinants of gait mechanics and the role of disease staging on these determinants. The study is quite interesting and I have just some important considerations (in addition to minor comments recorded below). The first issue is about the statistical model. Analyzing the results, especially looking qualitatively at the interactions between medication and group (fallers vs. non-fallers) it is clear that some outcomes (e.g., gait speed and stride length) were modified only in the faller group, clearly showing the role of medication more importantly in the faller group than in the non-faller group. The statistical model used prevents the strict analysis of this interaction, and an advanced General Linear Model is needed here (e.g. GLMM or GLMz). Another important point is about the possibility of analyzing walking speed in relative terms, controlling for size effects.

Particularly, I suggest trying to use one relatively new parameter to analyze the functional mobility. First, consider applying the rehabilitation locomotor index. To do this, you need just the walking speed and the lower limb length (great trochanter to the ground) or 0.54 of height (https://www.ncbi.nlm.nih.gov/pmc/articles/PMC2872302/)

After, you need to apply these two simple equations:

OWS (optimal walking speed, in m/s) = sqrt ( 0.25 x 9.81 x lower limb length (or 0.54 of height))

LRI (locomotor rehabilitation index, in %) = 100 x walking speed / OWS

The message is obtain the walking speed normalized based on theory of dynamic similarities and given an index that represents how is the person is close to your more economical metabolically to his/her optimal walking speed and where the pendular mechanism is more optimized. To understand in depth, please read: http://www.clinicaltdd.com/text.asp?2016/1/2/86/184750

Throughout the article, the authors use the term kinetics, when in fact, as far as I have reviewed, the correct term would be kinematics. Even when accelerometry is used, the measurements that are analyzed are kinematic. I missed detailed information on what and how the algorithm works to determine the events that make it possible to analyze the spatiotemporal parameters and information on possible filtering used on the kinematic data. Another important comment is about improving the discussion as the study extends the results of a recent meta-analysis on gait kinematics in Parkinson's, noting that medication seems to especially affect gait mechanics in people with falling PD (more than in non-falling PD). Explore these results further. Another important point is that the data related to the task of making the turn (changing direction) was not as determinant for fall risk as already found in the elderly. Discuss this point.

Minor points

Lines 102-103 – I mean kinematic instead of kinetic

Table 1 – body mass instead of body weight

Please take more care when revising the draft because some letter/numbers are chopped with the line number.

Author Response

Comments and Suggestions for Authors

Main comments

Dear authors, the topic of functional mobility and gait characteristics in people with lumbar canal stenosis is timely and interesting. The article aimed to compare gait kinematics between fallers and non-fallers, and with the effect of medication. In addition, regression models were performed in order to understand the determinants of gait mechanics and the role of disease staging on these determinants. The study is quite interesting and I have just some important considerations (in addition to minor comments recorded below).

  1. The first issue is about the statistical model. Analyzing the results, especially looking qualitatively at the interactions between medication and group (fallers vs. non-fallers) it is clear that some outcomes (e.g., gait speed and stride length) were modified only in the faller group, clearly showing the role of medication more importantly in the faller group than in the non-faller group. The statistical model used prevents the strict analysis of this interaction, and an advanced General Linear Model is needed here (e.g., GLMM or GLMZ).

Answers: Thanks for your comment. In this study, we added the following statistical analysis according to your comment. They are as follows:

The repeated-measures ANOVA was used to compare the parameter of locomotor rehabilitation index before and after dopamine replacement therapy between fallers and non-fallers during the straight forward and turning of walking cycle, with adjusting sex and age as potential confounding variables.

  1. Another important point is about the possibility of analyzing walking speed in relative terms, controlling for size effects.

Answers: Thanks for your comment. We calculate and listed the optimal walking speed between fallers and non-fallers according to your comment. We listed these data in newly created Table 3.

  1. Particularly, I suggest trying to use one relatively new parameter to analyze the functional mobility. First, consider applying the rehabilitation locomotor index. To do this, you need just the walking speed and the lower limb length (great trochanter to the ground) or 0.54 of height https://www.ncbi.nlm.nih.gov/pmc/articles/PMC2872302/) After, you need to apply these two simple equations:
    1. OWS (optimal walking speed, in m/s) = sqrt (0.25 x 9.81 x lower limb length (or 0.54 of height))
    2. LRI (locomotor rehabilitation index, in %) = 100 x walking speed / OWS

The message is obtaining the walking speed normalized based on theory of dynamic similarities and given an index that represents how is the person is close to your more economical metabolically to his/her optimal walking speed and where the pendular mechanism is more optimized. To understand in depth, please read: http://www.clinicaltdd.com/text.asp?2016/1/2/86/184750. Throughout the article, the authors use the term kinetics, when in fact, as far as I have reviewed, the correct term would be kinematics. Even when accelerometry is used, the measurements that are analyzed are kinematic. I missed detailed information on what and how the algorithm works to determine the events that make it possible to analyze the spatiotemporal parameters and information on possible filtering used on the kinematic data.

Answers: Thanks for your comment. We add the assessment of functional mobility in Patients and Method section according to your comment. They are as follows:

Assessment of the functional mobility

Optimal walking speed (OWS) is the walking speed adjusted for the lower limb length or height effects (OWS = sqrt (0.25 x 9.81 x lower limb length (or 0.54 of height)). In this situation, the speed is optimized and the metabolic energy expenditure is minimized.  Locomotor rehabilitation index (LRI), which expresses the relationship between the walking speed and OWS, is a complementary analysis of spatiotemporal parameters in gait assessment (LRI (%) = 100 x walking speed / OWS). We calculated OWS, and LRI at straight forward and turning during the walking cycle before and after dopamine replacement therapy between fallers and non-fallers groups.

  1. Another important comment is about improving the discussion as the study extends the results of a recent meta-analysis on gait kinematics in Parkinson's, noting that medication seems to especially affect gait mechanics in people with falling PD (more than in non-falling PD). Explore these results further.

Answers: Thanks for your comment. We added the following sentences in Discussion and try to make it clear. They are as follows:

Dopamine replacement therapy may improve speed and stride length, as well as decrease step length variability corresponding to increased hip ROM, ultimately making gait more efficient. Although levodopa has been shown to improve bradykinesia and rigidity, treatment with levodopa increases postural sway abnormalities in patients with PD. It also has shown a limited response to levodopa replacement therapy during the late stages of PD. Motor complications including dyskinesia, motor fluctuation, and off dystonia could occur in advanced PD. Once dopaminergic therapy has been started, off dystonia may appear when there is a decrease in brain dopamine levels while levodopa-induced dyskinesia may compromise balance and contribute to postural instability and falls. Our patient who had dyskinesia also had a higher risk of falls.

  1. Another important point is that the data related to the task of making the turn (changing direction) was not as determinant for fall risk as already found in the elderly. Discuss this point.

Answers: Thanks for your comment. Our study demonstrated that turning speed (m/sec) during the ‘off’ stage, but not the ‘on’ stage were significantly lower in those patients who had retrospective fallers group. Our study also highlighted the hypothesis that clinical disease severity mediates the relationship between turning speed and the risk of falling in patients with PD.  We added the following sentence in Discussion section and try to make it clearly. They are as follows:

Difficulty in turning is a well-known contributor to the risk of falling in older people. One study demonstrated that quality of turning, including mean turn duration, mean peak speed, and mean the number of steps to complete a turn, were was lower in recurrent-fallers compared to non-fallers and single-fallers. Our study demonstrated that turning speed during the ‘off-stage rather than the ‘on’ stage was significantly lower in those patients who had retrospective fallers group. In contrast to LRI in straight walking, LRI in turning walking did not show significant improvement after dopaminergic therapy. We also highlighted the hypothesis that clinical disease severity mediates the relationship between turning speed and the risk of falling in patients with PD.

  1. Minor points: Lines 102-103 – I mean kinematic instead of kinetic

Answers: Thanks for your comment. We correct “kinetic” into “kinematic” according to your comment.

  1. Table 1 – body mass instead of body weight. Please take more care when revising the draft because some letter/numbers are chopped with the line number.

Answers: Thanks for your comment. We correct “body weight” into “body mass” according to your comment.

Round 2

Reviewer 2 Report

The paper improved. I found out some minor points, but these points do not preclude the acceptation of article.

Line 295, 297, 299, 300, elsewhere - walking instead of waking

Perhaps in table 2 consider highlighting the subtitles (on/off phases).

  • consider letting s as abbreviation of speed, not sec (table 2, 5 and elsewhere)
  • minor minor comment: space before parentheses in table 2 and elsewhere.